# Differential Neural Correlates in the Prefrontal Cortex during a Delay Discounting Task in Healthy Adults: An fNIRS Study

**DOI:** 10.3390/brainsci13050758

**Published:** 2023-05-03

**Authors:** Masanaga Ikegami, Michiko Sorama

**Affiliations:** 1Department of Psychology, Asahikawa Medical University, Asahikawa 078-8510, Japan; 2Department of Psychology, Kyoto Notre Dame University, Kyoto 606-0847, Japan; msorama@notredame.ac.jp

**Keywords:** delay discounting, impulsivity, prefrontal cortex, functional near-infrared spectroscopy

## Abstract

The phenomenon of future rewards being devalued as a function of delay is referred to as delay discounting (DD). It is considered a measure of impulsivity, and steep DD characterizes psychiatric problems such as addictive disorders and attention deficit/hyperactivity disorder. This preliminarily study examined prefrontal hemodynamic activity using functional near-infrared spectroscopy (fNIRS) in healthy young adults performing a DD task. Prefrontal activity during a DD task with hypothetical monetary rewards was measured in 20 participants. A discounting rate (*k*-value) in the DD task was determined on the basis of a hyperbolic function. To validate the *k*-value, a DD questionnaire and the Barratt impulsiveness scale (BIS) were administered after fNIRS. The DD task induced a significant increase in oxygenated hemoglobin (oxy-Hb) concentration bilaterally in the frontal pole and dorsolateral prefrontal cortex (PFC) compared with a control task. Significant positive correlations were detected between left PFC activity and discounting parameters. Right frontal pole activity, however, showed significantly negative correlation with motor impulsivity as a BIS subscore. These results suggest that left and right PFCs have differential contributions when performing the DD task. The present findings suggest the idea that fNIRS measurement of prefrontal hemodynamic activity can be useful for understanding the neural mechanisms underlying DD and is applicable for assessing PFC function among psychiatric patients with impulsivity-related problems.

## 1. Introduction

Everyone has experience in yielding to the temptation of going back to sleep instead of choosing healthy habits such as getting up early. The former can provide immediate gratification, whereas the latter results in larger benefits in the future. With such choices, future outcomes are devalued as a function of the temporal delay and we therefore often choose smaller, more immediate rewards. This phenomenon is referred to as delay discounting (DD) and is considered a measure of impulsivity [1]. Conversely, choosing a larger but delayed reward is defined as self-control [2]. A higher tendency to discount future rewards is considered to have negative effects on daily activities and health. For example, a tendency to place less weight on future outcomes is associated with lower academic performance among college students [3] and a higher degree of DD represents a significant risk factor for becoming overweight or obese [4].

In terms of clinical perspective, steep DD or the tendency to discount future rewards more rapidly has been shown to characterize psychiatric problems such as substance-related and addictive disorders (e.g., alcohol use disorder and pathological gambling) [5,6] and neurodevelopmental disorders including attention deficit/hyperactivity disorder (ADHD) [7,8] and autism spectrum disorders (ASD) [9]. For example, children with ADHD exhibit a strong bias towards small, immediate rewards over large, delayed rewards more frequently than typically developing (TD) children [10]. Ikegami et al. [11] reported that Japanese children with ADHD and children with ASD showed steep discounting of monetary reward compared to TD children, and the discounting rate correlated with impulsive choice tendencies in daily life. Recent meta-analyses have concluded that children with ADHD exhibit significantly increased discounting compared to TD children [7,8], suggesting that steep discounting contributes to the diagnostic features of ADHD as well as a weakness of inhibitory control [12].

Recent neuroimaging studies mainly using functional magnetic resonance imaging (fMRI) have investigated the neural mechanisms underlying DD. Peters and Büchel [13] outlined that individual differences in DD are mediated by multiple neural processes involved in reward valuation, cognitive control of decision conflict, and imagery of future outcomes. First, neural activities in the brain reward system, which mainly consists of the ventral striatum and ventromedial prefrontal cortex (PFC), correlate with the subjective value of delayed rewards measured behaviorally [14]. Ballard and Knutson [15] revealed that more discounting individuals showed less striatal activation to delayed rewards, suggesting that lower reactivity in the brain reward system is related to impulsive choice behaviors. Second, the lateral PFC also plays an important role in the valuation of future outcomes, particularly when the individual is forced to make a difficult choice under conditions of temptation [16]. More specifically, the lateral PFC is proposed to contribute to self-control choices by incorporating long-term considerations into value signals encoded in the ventromedial PFC [17]. Reduced activation of the lateral PFC can thus bring about impulsive choice behaviors [18]. Interestingly, the characteristic of steep discounter shown in individuals with impulse control disorders may also be linked to lateral PFC deactivation during executive function tasks demanding working memory and response inhibition, implying that the neurobiological substrates of the various executive functions and impulse control overlap within the lateral PFC [19,20]. Lastly, the medial temporal lobe network, including the hippocampus, is involved in vivid imagery of future outcomes, and activities of the network promote self-control behaviors [21,22].

In these multiple neural processes, attention has focused on the lateral PFC as a target of clinical interventions, including cognitive training, pharmacological treatments, and transcranial direct current stimulation to improve impulsivity [23,24,25]. Similarly, the measurement of cortical activity associated with DD may be available for neurological assessment of impulse control problems such as ADHD [26].

Recent advances in techniques for monitoring cortical hemodynamics using functional near-infrared spectroscopy (fNIRS) have enabled the evaluation of neurocognitive functions in diverse participants, including children with ADHD, e.g., [27]. Compared to fMRI, fNIRS has the advantages of being complete noninvasive and showing relative insensitivity towards movement-related artifacts, and is well suited for rapid measurement due to the easy application. We therefore believe that fNIRS measurement of prefrontal hemodynamic activity associated with DD could offer a useful approach by which to assess lateral PFC functions in patients with impulsivity-related problems. However, to the best of our knowledge, no fNIRS studies have reported the prefrontal activity associated with DD, in either clinical groups or healthy populations.

In the present study, we aimed to conduct a preliminary examination of prefrontal hemodynamic activity in healthy young adults performing a DD task using multichannel fNIRS. A DD task is a one of the most popular paradigms for evaluation of DD through choices between smaller, relatively immediate and larger, more delayed rewards [28]. We adapted the task to simultaneous fNIRS measurement using a simplified method with a smaller number of choice trials. Based on previous fMRI studies, we hypothesized that significant prefrontal activation would be observed during the DD task compared with a control choice task, in which participants were offered a choice between two rewards that differed only in amount or delay. In addition, we explored relationships between the lateral PFC activities during the DD task, discounting parameters, and trait impulsivity as assessed using a self-reported measure.

## 2. Materials and Methods

### 2.1. Participants

A total of 20 healthy young adults (8 males, 12 females; mean age = 24.95 years; standard deviation = 4.95 years) participated in this study as paid volunteers. Participants were paid 1000 yen per hour after the experiments. The handedness of each participant was assessed using the Edinburgh Handedness Inventory [29] and all except for three male participants were right-handed. None of the participants had any history of psychiatric/neurological disorders. Written informed consent was obtained from all participants prior to the experiments. This study complied with the Declaration of Helsinki and was approved by the ethics committee of Asahikawa Medical University (approval no. 595-2).

### 2.2. DD Task

All experiments were conducted in a small, sound-shielded, dimly lit room containing a small desk and a chair. Participants sat at a table on which a notebook computer (Latitude E6500; DELL, Round Rock, TX, USA) was placed to present visual stimuli. Viewing distance was held constant at approximately 50 cm. A behavioral task was created using Visual Basic 6.0 (Microsoft, Redmond, WA, USA). Participants responded to stimuli using a mouse connected to the computer.

Participants performed a computerized DD task with a hypothetical monetary reward during fNIRS measurement. The task was modelled on the monetary choice questionnaire to estimate the individual discounting rate through 21 choices between smaller immediate and larger delayed rewards [28,30]. We adapted the task to simultaneous fNIRS measurement with a block design using a smaller number of choice trials. Specifically, participants were asked to make repeated choices between a smaller reward (70,000 yen) delivered immediately and a larger reward (90,000 yen) delivered after a variable delay (tomorrow, 1 week, 2 weeks, 1 month, 3 months, 6 months, 1 year, 2 years, 5 years, and 10 years), resulting in 10 pairs of choices in the task. Figure 1a (left panel) provides a representative example of the DD task. In the trial, the left option shows a small, immediate reward and the right option shows large reward delivered in 3 months. Left or right position of the delayed reward was counterbalanced across trials. Each delay was presented in random order among participants.

Participants also performed a control task in which they repeatedly chose between two options comprising different reward amounts delivered immediately (10,000 yen vs. 20,000 yen; 10,000 yen vs. 30,000 yen; or 20,000 yen vs. 50,000 yen) or between two options comprising a fixed reward amount (30,000 yen) delivered after different delays (2 weeks or 1 year), resulting in five binary choices in the task. Figure 1a (right panel) illustrates an example trial of the control task. These choices were presented in random order among participants. In contrast to the DD task, the control task required participants to compare only one dimension, either the relative amount of reward or the delay, between alternatives [31,32]. The control task was utilized as a baseline against which to determine relative changes in fNIRS signals during the DD task.

As shown in Figure 1b, control and DD tasks were administered alternately. Each DD task was divided into 2 blocks of 5 trials each to correspond to the length of an fNIRS measurement block (30 s), and two sets of the DD task were performed overall. Each block consisted of 5 control trials and 5 DD trials, with 5 control trials added after the last block. As a result, each participant performed a total of 25 control trials and 20 DD trials overall. In each trial, a pair of options was displayed for 4 s and participants were allowed to make a choice within 6 s immediately after the presentation of options. Participants chose an option by pressing the corresponding left or right mouse button, then options disappeared. Reaction time and choice were recorded for each trial of both tasks.

The discounting rate in the DD task was determined on the basis of a hyperbolic function, V = A/(1 + *k*D), where V is the subjective value of the delayed reward, A is the amount of the reward, D is the length of delay, and *k* is a free parameter representing the discounting rate [31]. According to the procedure described by Kirby and Maraković [28], for each pair of options, we calculated the value of the discounting rate parameter (*k*) for which the subjective value of the delayed reward is equal to that of the immediate reward (see Table 1). An actual indifference point for each participant was assumed to exist between the point of switching from delayed reward to immediate reward and immediately before that switching point. Therefore, individual *k*-values were calculated as the geometric mean between *k*-values at and immediately before the switching point. Finally, the geometric mean between *k*-values obtained from the first set (2 blocks in the first half) and second set (2 blocks in the latter half) was defined as the discounting rate for each participant. Because discounting rates did not follow a normal distribution and were positively skewed by low-frequency, high-value outliers [33], we performed common logarithmic transformation of *k*-values in subsequent analyses.

### 2.3. Questionnaires to Assess Impulsivity

To validate *k*-values during fNIRS measurement, we also used a standard DD questionnaire in which participants were asked to make repeated choices between a small immediate reward and a larger reward with variable delays [30,34]. The questionnaire consisted of five conditions representing different delays (6 months, 1 year, 5 years, 10 years, and 20 years) for a fixed hypothetical reward of 100,000 yen. Each delay condition included 25 pairs of choices: The left options showed immediate rewards changing within the range of 1000 yen to 99,000 yen in 25 steps and the right options showed a fixed reward of 100,000 yen with a delay. Immediate reward amounts in each delay condition were presented in ascending and descending orders. Participants completing the fNIRS measurement received a booklet printed with 25 pairs of choices per sheet. After brief instruction, the participant repeatedly chose between small immediate and large delayed rewards at their own pace. Indifference points for each delay condition were calculated by averaging the first immediate amount chosen in the ascending series and the last immediate amount chosen in the descending series [35]. According to the procedure described by Myerson et al. [33], as a measure of delay discounting, the area under the curve (AUC) was calculated with the indifference points of each participant. First, delays and subjective values were normalized by expressing delays as proportions of the maximum delay (20 years) and expressing subjective values as proportions of the amount of the delayed reward (100,000 yen). These normalized values were used as x coordinates and y coordinates, respectively, to construct the discounting function. This AUC represents the area under the empirical discounting function produced by connecting indifference points under each delay condition with straight lines. Thus, a smaller AUC reflects a higher degree of discounting.

Finally, participants completed the Japanese version of the Barratt impulsiveness scale (BIS) 11th version [36] to assess trait impulsivity. The BIS is a self-reported measure consisting of 30 items that assess three sub-components revealed by the factor analysis of impulsive personality traits: attentional impulsivity (Iat), showing lack of attention to details; motor impulsivity (Im), showing action without thinking; and non-planning impulsivity (Inp), showing present orientation or lack of futuring [37].

### 2.4. fNIRS Data Acquisition

According to the procedure described by Takahashi and Ikegami [38], temporal changes in concentrations of oxygenated hemoglobin (oxy-Hb) and deoxygenated hemoglobin (deoxy-Hb) during the DD task were measured using a 24-channel fNIRS instrument (ETG-100; Hitachi Medical Corporation, Tokyo, Japan) at two wavelengths of near-infrared light (780 and 830 nm). The distance between the light emitter and detector probes was set at 30 mm to detect hemodynamic changes at the surface of the cerebral cortex, 20–30 mm below the scalp [39]. Although oxy- and deoxy-Hb concentration changes depend on the optical pathlength of near-infrared light in brain tissue, the optical pathlength cannot be referenced in Hb concentration measurements using a continuous wave laser. Therefore, the oxy-Hb and deoxy-Hb signals were calculated in units of millimolar × millimeter (mM × mm). Data were measured at a sampling rate of 10 Hz.

FNIRS probes were placed on the forehead of the participant to cover the lateral PFC of both hemispheres. Two sets of 3 × 3 probe holders, each of which included 12 measurement channels, were placed symmetrically on the left and right frontal area by referring to the international 10–20 system used in electroencephalography. Positions of the probes and reference points (nasion, midline central, and preauricular points) on the scalp were recorded using a three-dimensional digitizer (Isotrack II, Polhemus, Colchester, VT, USA). As shown in Figure 2a, the lowest probes in left and right medial columns were positioned at Fp1 and Fp2, respectively. Based on probe positions for each participant, spatial information for each channel was estimated using a probabilistic registration method [40] to register channel positions to the Montreal Neurological Institute (MNI) standard brain space [41]. Figure 2b represents estimated channel positions on the MNI brain, and indicates that the measurement area included the frontal pole and dorsolateral and ventrolateral PFC bilaterally (Brodmann areas 9, 10, 45, 46).

The ETG-100 system measured change in the concentrations of oxy-Hb and deoxy-Hb from the starting baseline. Individual time course data for oxy-Hb and deoxy-Hb concentrations were averaged over the four blocks. As a preprocessing step for individual raw data of oxy-Hb and deoxy-Hb, a moving average method was employed to remove high-frequency fluctuations presumably caused by pulsation (moving average window, 5 s). To correct for low-frequency spontaneous fluctuations, baseline correction was performed using linear fitting based on two baseline periods: the pre-task baseline, determined as the mean across a 5 s period just before the onset of the DD task period; and the post-task baseline, determined as the mean across a 5 s period 20 s after the DD task period, using the integral mode installed in ETG-100.

### 2.5. Data Analyses

Mean reaction time in the DD task was compared to that in the control task using the paired *t*-test. Relationships between behavioral parameters (reaction time and *k*-value) and indicators of impulsivity measured by questionnaires (AUC and BIS subscores) were examined using Pearson’s correlation coefficient.

For statistical analyses of changes in the concentrations of oxy-Hb and deoxy-Hb, mean values of the pre-task baseline and task period were calculated for each participant and fNIRS channel. Of the 24 channels, 10 located in the upper area of the probes (left: CH1–5 and right: CH13–17) were affected by measurement noise because of poor optical contact due to high hair density of some participants [42], so those channels were excluded from further analyses. Hemoglobin concentrations were analyzed using a two-way repeated-measures analysis of variance (ANOVA) with channel (14 channels) and time segment (pre-task baseline vs. task period) as within-subject factors, followed by post hoc analysis for simple main effects. Although the optical pathlength in fNIRS varies among individuals, this is not an issue when comparing the same channel in the same individual across different task conditions. Therefore, differential pathlength factor was not defined in the present study. FNIRS channels showing significant activation during the DD task, as determined by two-way ANOVA, were selected for further analyses to examine the relationships between oxy-Hb concentration and impulsivity measures (*k*-value, AUC, and subscores of the BIS) using Pearson’s correlation coefficient.

## 3. Results

### 3.1. Behavioral Performances and Questionnaires

Table 2 summarizes the results of behavioral performances and questionnaires. Paired *t*-test revealed that reaction time was significantly longer during the DD task than during the control task (*t* (19) = −4.678, *p* < 0.001, Cohen’s *d* = −1.046). The *k*-value obtained from the DD task correlated negatively with the AUC assessed by the discounting questionnaire (*r* = −0.488, *p* = 0.029). BIS-Im score correlated negatively with reaction times in the control task and in the DD task (*r* = −0.564, *p* = 0.010 and *r* = −0.473, *p* = 0.035, respectively). No significant correlations were identified between BIS subscores and either *k*-value or AUC. The correlation coefficients between these behavioral measures are summarized in Table 3. Figure 3 displays two scatter plots showing the negative correlation between the AUC and the *k*-value, and the negative correlation between mean reaction time in the DD task and the BIS-Im score, respectively.

### 3.2. fNIRS Data

Figure 4a shows mean relative changes in oxy-Hb and deoxy-Hb concentrations during the task period (the DD task) relative to the pre-task baseline (the control task) value. As a whole, performing the DD task elicited significant activation in the frontal pole and dorsolateral PFC bilaterally (BA10/46), indicated by increases in oxy-Hb concentration. Two-way repeated-measures ANOVA revealed a significant main effect of channel (F(13, 247) = 2.298, *p* = 0.007, η_p_^2^ = 0.108) and time segment (F(1, 19) = 13.950, *p* = 0.001, η_p_^2^ = 0.423) on oxy-Hb concentration. In addition, a significant interaction was seen for channel × time segment (F(13, 247) = 2.180, *p* = 0.011, η_p_^2^ = 0.103). To examine whether oxy-Hb concentration in each channel changed significantly during the task period relative to the pre-task baseline, simple main effects of the time segment factor for each channel were analyzed. The result indicated that oxy-Hb concentration was higher in the task period than in the pre-task baseline period in three channels over the left PFC (CH8, CH9, and CH11; F(1, 19) = 16.515, *p* < 0.001, η_p_^2^ = 0.465, F(1, 19) = 7.080, *p* = 0.015, η_p_^2^ = 0.272, and F(1, 19) = 11.039, *p* = 0.004, η_p_^2^ = 0.368, respectively) and five channels over the right PFC (CH18, CH19, CH21, CH22, and CH24; F(1, 19) = 17.623, *p* < 0.001, η_p_^2^ = 0.481, F(1, 19) = 5.716, *p* = 0.027, η_p_^2^ = 0.231, F(1, 19) = 15.085, *p* = 0.001, η_p_^2^ = 0.443, F(1, 19) = 17.234, *p* < 0.001, η_p_^2^ = 0.476, and F(1, 19) = 10.805, *p* = 0.004, η_p_^2^ = 0.363, respectively). Table 4 shows estimated locations of these activated channels in MNI coordinates. For deoxy-Hb concentration, two-way repeated-measures ANOVA revealed a significant main effect of time segment (F(1, 19) = 6.780, *p* = 0.017, η_p_^2^ = 0.263), indicating smaller deoxy-Hb concentration in the task period than in the pre-task baseline period. No significant interaction was seen for channel × time segment for deoxy-Hb concentration. Figure 4b shows grand-averaged wave forms in representative channels on the left (CH8) and right (CH22) PFC.

We examined the relationship between activating effects observed in eight channels and impulsivity measures (*k*-value, AUC, and subscores of the BIS). The increased oxy-Hb concentration in CH8, CH9, CH11, and CH24 were correlated with larger *k*-value (*r* = 0.477, *p* = 0.034, *r* = 0.528, *p* = 0.017, *r* = 0.459, *p* = 0.042, and *r* = 0.492, *p* = 0.028, respectively). In addition, the increased oxy-Hb concentration in CH9 was correlated with a smaller AUC (*r* = −0.468, *p* = 0.038). Likewise, the increased oxy-Hb concentration in CH9 was correlated with the larger BIS-Inp score (*r* = 0.464, *p* = 0.039). On the other hand, the increased oxy-Hb concentration in CH22 was correlated negatively with the smaller BIS-Im score (*r* = −0.557, *p* = 0.011). There was no significant correlation between oxy-Hb concentration and the BIS-Iat score. Figure 5 depicts two scatter plots that illustrate the positive correlation between oxy-Hb concentration in the left dorsolateral PFC (CH9) and the *k*-value, as well as the negative correlation between oxy-Hb concentration in the right frontal pole (CH22) and the BIS-Im score.

## 4. Discussion

The present study investigated prefrontal hemodynamic activity in healthy young adults performing the DD task using a multichannel fNIRS system. To examine an individual *k*-value as a discounting rate, we used a simplified method with a smaller number of choices during simultaneous fNIRS measurement. Participants with larger *k*-values showed smaller AUC in the standard DD questionnaire, indicating that the *k*-value offered a valid measure of DD. In addition, the DD task significantly prolonged the mean reaction time for choice responses compared to that of the control task. These results suggest that the DD task successfully elicited decision-making processes relevant to the trade-off between amounts and delays.

On the other hand, BIS subscores did not correlate significantly with *k*-value or AUC. Previous studies have reported that relationships between discounting parameters and self-reported measures, including the BIS, were either not significant or significant but weak [30,43,44]. For example, Saeki [30] reported that the *k*-value obtained using a method similar to that of the present study correlated significantly but weakly with BIS-Inp (non-planning impulsivity) score in 116 university students (*r* = 0.23, *p* < 0.05). In accordance with these studies, we confirmed the same level of correlation coefficient between BIS-Inp and *k*-value (*r* = 0.215), although it was not significant due to smaller sample size. Given that DD of monetary rewards may be affected by state-dependent factors including socioeconomic status such as income level [45,46] and physiological status such as hunger [47] as well as trait impulsivity, further research with a larger sample size and controlling these factors is needed to clarify the relationship between discounting parameters measured behaviorally and self-reported measures of impulsivity.

We determined that performing the DD task produced increases in oxy-Hb concentration in the bilateral frontopolar area (BA10) and dorsolateral PFC (BA45/46) compared to performing the control task. Hemodynamic changes were consistent with previous fMRI studies showing engagement of the frontal pole and dorsolateral PFC in the DD task [16,48,49,50,51]. Lateral PFC activation likely reflects specific aspects of the DD task, in which participants are required to make trade-off decisions between reward amounts and delays [31,32]. In such choices, working memory has been suggested to play an important role in actively maintaining reward values as diverse information is manipulated and integrated to choose between alternatives that differ in amounts and delivery timing [52]. In fact, imposing a working memory load leads to greater discounting of delayed rewards [53], implying that performing the DD task requires working memory. This hypothesis is further supported by the finding of a functional overlap between working memory and DD, particularly in the left dorsolateral PFC [20]. Therefore, the dorsolateral PFC activity observed in the present study can be plausibly assumed to be associated with working memory processes relevant to the task demand of the DD task. In addition, frontal pole activity might be associated with a process of imagining future rewards in the DD task [48]. This speculation is consistent with findings that the rostral prefrontal cortex (BA10) supports farsighted decisions by increasing functional coupling with the hippocampus, which is assumed to be critical for episodic future thinking [21,22]. Taken together, the significant activity observed in the frontopolar and lateral PFC during the DD task supports previous theories proposing that these cortical regions integrate reward-related information and influence goal-directed behavior. It is known that these lateral PFC regions, as well as the ventromedial PFC, project to the ventral striatum, the center of the brain reward system [54]. That is, these prefrontal regions are thought to be part of an executive system that modulates the striatal reward circuit to guide choice behavior [17,23].

In addition, we detected medium to large positive correlations between hemodynamic changes in the predominantly left PFC and discounting parameters (*k*-value and AUC), indicating that individuals with a higher degree of DD showed more PFC activation during the DD task. Similarly, higher BIS-Inp score was moderately associated with the greater activity of the left PFC. These results are somewhat surprising given the fact that inhibition of the left PFC can lead to impulsive choices [18]. However, similar to the present results, several reports have shown paradoxical relationships in healthy adults, with individuals with a higher degree of DD having the greater PFC activity in the potentially real DD task [49] and the hypothetical DD task [55]. The positive correlations between self-rated impulsivity or *k*-value and PFC activity performing the DD task were also obtained in a mixed population of healthy and addicted individuals [32,56]. A possible explanation of these results would be that individuals with a higher degree of DD required more processing resources to deal with the trade-off between reward amount and delay, and for individuals with a lower degree of DD, this task might be cognitively less demanding. This is consistent with the idea that individuals who prefer immediate rewards do so because of inefficient cognitive control, and they need to activate the cognitive circuit more strongly to make a delayed choice [32]. From another perspective, for individuals with low levels of impatience towards money, intertemporal choices may seem merely a matter of preference rather than a matter of self-control [49]. To confirm this hypothesis, additional studies controlling participants’ income levels and using tasks with tailored choice difficulty will be needed.

Interestingly, we detected a large negative correlation between oxy-Hb concentration in CH22 located over the right frontal pole and the BIS-Im subscore. BIS-Im (motor impulsivity) measures a rough-and-ready decision style and lack of perseverance [37], and has been thought to reflect a similar concept of inhibitory control [57]. In fact, even in the present study, shorter reaction times correlated with higher BIS-Im scores. Our result suggests that individuals with lower inhibitory control show reduced activity of the right frontal pole during intertemporal decision-making. This interpretation aligns with previous fMRI studies showing negative correlations between right lateral PFC activity and BIS motor impulsivity [58] and a meta-analysis showing that response inhibition and DD both engage the right lateral PFC [20]. Likewise, recent revisions of the approach/avoidance motivational systems theory have shown that the right frontal area is a regulatory system that governs the two motivational systems and is responsible for a variety of inhibitory controls [59]. In the present study, the BIS-Im scores were not significantly correlated with the discounting parameters themselves, but they were related to shorter reaction times in the tasks. Therefore, it is possible that the right frontopolar activity is associated with response inhibition during the motor output phase of choice behavior. A recent neuroanatomical study, however, revealed that gray matter volume in the right frontal pole was predictive of DD, but not response inhibition, which was associated with the right inferior frontal gyrus [60]. This suggests neural dissociations between choice impulsivity and motor impulsivity within the right lateral PFC. Impulsive choice and impulsive action have been shown to be common characteristics of psychiatric problems such as addictive disorders [61] and ADHD [62]. Clearly, further exploration is necessary to provide a detailed map regarding neural substrates underlying DD and motor impulsivity.

Some methodological limitations to the present study should be considered. First, fNIRS measurement used a block design to compare prefrontal hemodynamic activities during the DD task with those during the control task. We therefore could not clarify differential cortical activities between impulsive choice trials and self-control trials based on each single choice. To establish the relationship between prefrontal activity and each type of choice, future studies should adopt an event-related design. A second limitation of the present study was that the sample size did not allow statistical analyses to be performed according to the socioeconomic status of participants such as income level. Because income level has been determined to correlate with DD as well as other executive functions [46], controlling for such potential confounding factors is important to investigate related prefrontal activities. Additional studies with a larger sample size capable of controlling for income level are needed to address this issue.

## 5. Conclusions

In conclusion, this is, to the best of our knowledge, the first fNIRS study to reveal prefrontal hemodynamic changes during the DD task in healthy young adults. We discovered positive correlations between left dorsolateral PFC activity and discounting parameters, and a negative correlation between right frontal pole activity and trait impulsivity related to inhibitory control. These results provide support for the idea that left and right PFCs have differential contributions when performing the DD task. The current results suggest that fNIRS measurement of prefrontal hemodynamic activity may be useful for understanding the neural mechanisms underlying DD and have applications in assessing lateral PFC function for psychiatric patients with impulsivity-related problems.

## Figures and Tables

**Figure 1 brainsci-13-00758-f001:**
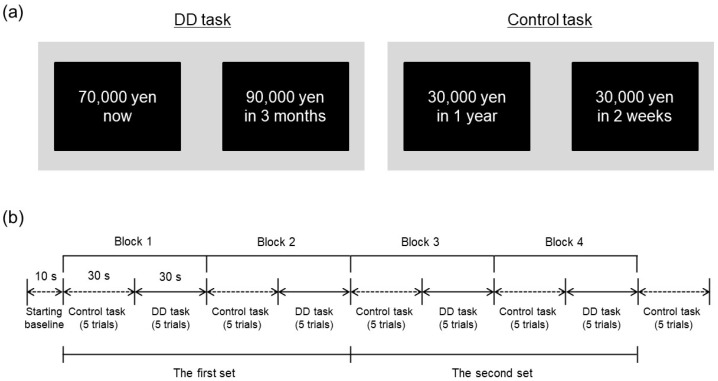
(**a**) Example trials of the DD task and control task. The left panel shows an example trial of the DD task. The right panel shows example trial of the control task, which presents two options with consisted fixed amounts of reward delivered after different delays. In the experiment, the options were presented in Japanese. (**b**) Time course of experimental session. FNIRS measurements were conducted from the starting baseline to the end of control trials after the last block.

**Figure 2 brainsci-13-00758-f002:**
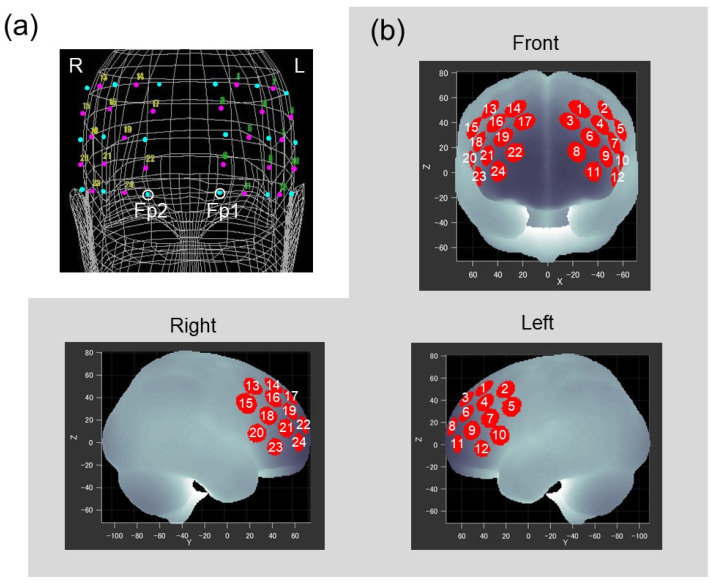
(**a**) Positions of the probes and channels as situated on the head of the representative participant. The two probe holders, each equipped with five laser diodes and four photodiodes (cyan dots), were located on the prefrontal regions. Open circles correspond to Fp1 and Fp2. Magenta dots with numbers represent the 24 measurement channels. (**b**) Channel positions registered to the MNI standard brain space. Red circles represent statistically estimated channel locations. White numbers indicate corresponding channels.

**Figure 3 brainsci-13-00758-f003:**
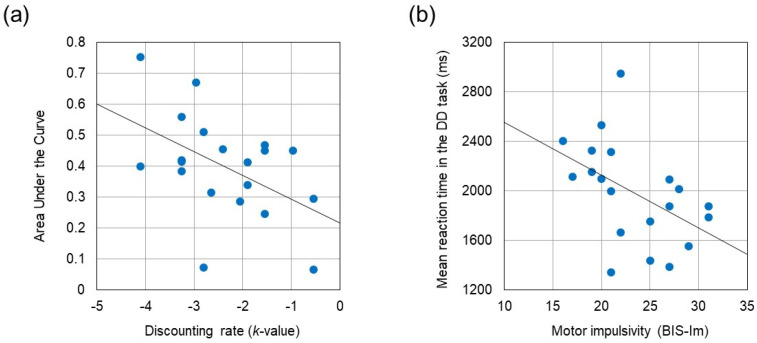
(**a**) Negative correlation between Area Under the Curve and the discounting rate (*k*-value) (*r* = −0.488, *p* = 0.029). (**b**) Negative correlation between mean reaction time in the DD task and the motor impulsivity (BIS-Im) (*r* = −0.473, *p* = 0.035).

**Figure 4 brainsci-13-00758-f004:**
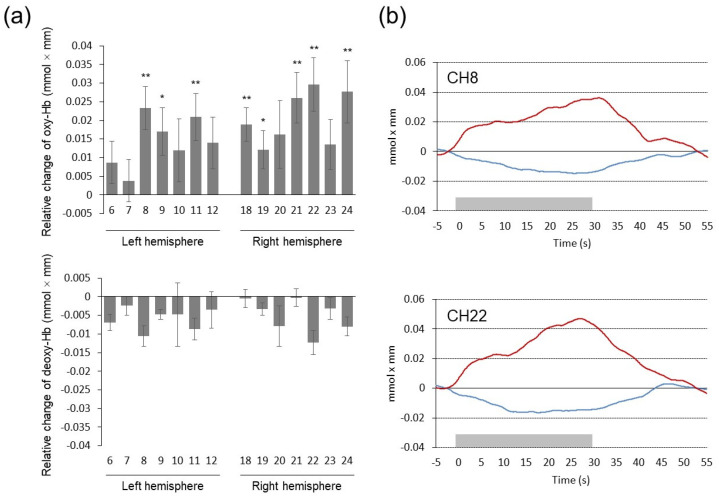
(**a**) Mean changes in oxy- and deoxy-Hb concentrations during the DD task relative to the control task. Numbers on the *x*-axis represent the measurement channel. Error bars represent standard errors. * *p* < 0.05, ** *p* < 0.01. (**b**) Grand-averaged waveforms of oxy- and deoxy-Hb concentrations in representative channels. Red lines represent oxy-Hb changes and blue lines represent deoxy-Hb changes. Gray rectangles on the *x*-axis represent task period.

**Figure 5 brainsci-13-00758-f005:**
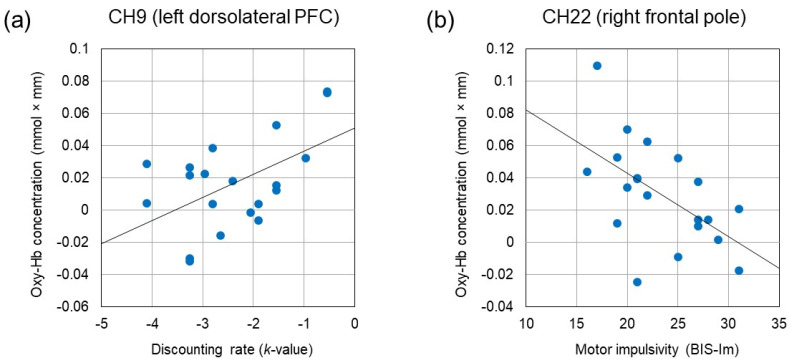
(**a**) Positive correlation between oxy-Hb concentration in the left dorsolateral PFC (CH9) and the discounting rate (*k*-value) (*r* = 0.528, *p* = 0.017). (**b**) Negative correlation between oxy-Hb concentration in the right frontal pole (CH22) and the motor impulsivity (BIS-Im) (*r* = −0.557, *p* = 0.011).

**Table 1 brainsci-13-00758-t001:** Choice trials and their associated discounting parameter values.

Trial No.	Delay	*k*-Value
1	1 day	0.28571
2	1 week	0.04082
3	2 weeks	0.02041
4	1 month	0.00952
5	3 months	0.00317
6	6 months	0.00159
7	1 year	0.00078
8	2 years	0.00039
9	5 years	0.00016
10	10 years	0.00008

Delay is the length of delay for larger reward. The discounting parameters (*k*-value) are those values at which the immediate and delayed rewards are of equal value according to the hyperbolic function.

**Table 2 brainsci-13-00758-t002:** Behavioral performances and questionnaires.

	Behavioral Performances		Questionnaires			
	RT (ms)	*k*-Value	AUC	BIS-Iat	BIS-Im	BIS-Inp
DD task	1984.0 ± 408.1 **	−2.37 ± 1.06	0.40 ± 0.17	16.85 ± 3.84	23.40 ± 4.54	24.70 ± 3.99
Control task	1782.0 ± 307.9					

Values are given as means ± SD. ** *p* < 0.01. The *k*-value is represented as the common logarithmic transformation. AUC is the abbreviation for the Area Under the Curve. BIS is the abbreviation for Barratt Impulsiveness Scale. Iat, attentional impulsivity; Im, motor impulsivity; Inp, non-planning impulsivity.

**Table 3 brainsci-13-00758-t003:** Correlation coefficients between behavioral measures.

	RT (Control)	RT (DD)	*k*-Value	AUC	BIS-Iat	BIS-Im	BIS-Inp
RT (control)	—	0.892 **	−0.048	0.265	−0.298	−0.564 **	−0.349
RT (DD)		—	−0.139	0.332	−0.165	−0.473 *	−0.311
*k*-value			—	−0.488 *	−0.142	0.075	0.215
AUC				—	0.342	0.072	−0.234
BIS-Iat					—	0.469 *	0.083
BIS-Im						—	0.537 *
BIS-Inp							—

* *p* < 0.05, ** *p* < 0.01. AUC is the abbreviation for the Area Under the Curve. BIS is the abbreviation for Barratt Impulsiveness Scale. Iat, attentional impulsivity; Im, motor impulsivity; Inp, non-planning impulsivity.

**Table 4 brainsci-13-00758-t004:** Most likely estimated locations of activated channels using probabilistic spatial registration.

	MNI Coordinate				Brodmann Area Estimation	
	X	Y	Z	SD (mm)	Brodmann Areas	%
CH8	−23.33	69.00	16.33	8.79	10—Frontopolar area	100.00
CH9	−46.33	50.67	12.33	9.36	46—Dorsolateral prefrontal cortex	75.55
CH11	−36.67	64.00	0.67	8.41	10—Frontopolar area	86.60
CH18	54.33	36.33	24.33	9.04	45—Pars triangularis Broca’s area	100.00
CH19	35.33	56.33	27.67	9.10	46—Dorsolateral prefrontal cortex	98.43
CH21	48.00	52.33	13.33	8.86	46—Dorsolateral prefrontal cortex	91.30
CH22	26.33	68.67	15.67	8.56	10—Frontopolar area	100.00
CH24	39.67	64.33	0.67	8.52	10—Frontopolar area	82.90

## Data Availability

The data that support the findings of this study are available from the corresponding author upon reasonable request.

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
