# Peer review of "Differential Neural Correlates in the Prefrontal Cortex during a Delay Discounting Task in Healthy Adults: An fNIRS Study"

_brainsci, 2023, doi:10.3390/brainsci13050758_

Round 1

Reviewer 1 Report

Comments and Suggestions for Authors

The study investigates the neural correlates of the prefrontal cortex in delay discounting through functional Near-Infrared Spectroscopy. The topic is interesting, and the paper is well written, but, in my opinion, some concerns need to be addressed:

1)     Please consider to refer to the NIRS as functional NIRS (fNIRS) since it is used to assess the cortical function. In my opinion, NIRS should be replaced with fNIRS in the manuscript.

2)     In my opinion the statistical analysis should be better explained. It is not clear to me whether the post-hoc comparisons of the ANOVA were corrected for multiple comparisons.

3)     Please add some more information regarding the fNIRS preprocessing: Have the signals been filtered? If yes, please define the kind of filter employed and the cut-off frequencies. How the differential pathlength factor was defined for the participants?

4)     In the discussion section please provide some comments regarding the entity of the correlations found referring to, for instance, the correlation coefficients interpretation proposed by Cohen. 

Reviewer 2 Report

Comments and Suggestions for Authors

This study aimed to investigate the effects of a delay discounting task on PFC hemodynamic activity using NIRS. The authors also explored relationships between lateral PFC activities, discounting parameters, and trait impulsivity during this task. Overall, the manuscript is well-structured and well-written. Therefore, I would recommend the publication of this manuscript. Before that though, some aspects need to be further discussed. The following are suggestions and issues that the authors should address.

· Introduction: Are there any differences between the terms “Delay discounting”, “Higher discounting”, and “Steep discounting”? As a reader, I got a bit confused when I read the first two paragraphs of the introduction. If the above terms are the same, please either mention it somewhere at the beginning of the introduction or choose only one of these terms and use it throughout the whole manuscript in order to keep consistency. If there are some differences between “Delay discounting”, “Higher discounting”, and “Steep discounting”, please clearly explain them in the introduction.

· Results: As a reader, I would like to see some results of this study in a figure or a table. In the first paragraph of the Results section, the authors mentioned, "The k-value obtained from the DD task correlated negatively with the AUC assessed by the discounting questionnaire (r = -0.488, p = 0.029). BIS-Im score correlated negatively with reaction times in the control task and in the DD task (r = -0.564, p = 0.010 and r = -0.473, p =0.035, respectively). No significant correlations were identified between BIS subscores and either k-value or AUC". Why didn't the authors display these excellent results in a figure? Like Figure 4, they could have also shown these results with scatter plots.

· Discussion: Please shortly discuss the “appetitive/approach motivation” and “withdrawal/avoidance motivation” models in the discussion section, as they are associated with asymmetry of the frontal regions. Have a look at the following papers.

[1]. P. A. Gable, L. B. Neal, and A. H. Threadgill, “Regulatory behavior and frontal activity: considering the role of revised-BIS in relative right frontal asymmetry,” Psychophysiology 55, e12910 (2018).

[2]. B. D. Nelson et al., “Depression symptom dimensions and asymmetrical frontal cortical activity while anticipating reward,” Psychophysiology 55, e12892 (2018).

[3] Zohdi H, Scholkmann F, Wolf U. Frontal cerebral oxygenation asymmetry: intersubject variability and dependence on systemic physiology, season, and time of day. Neurophotonics. 2020;7(2):025006.

Reviewer 3 Report

Comments and Suggestions for Authors

This is an interesting study that identified neural correlates of delay discounting in the prefrontal areas using NIRS. Overall, the paper is well written and is easy to read. Statistical analyses are good, but please revise the mathematical symbol (subscript and superscript) of the effect sizes, partial eta-squared.

1) The word "Left and Right" before 'prefrontal' in the title feels redundant as there are only two sides of the brain. In fact, I would suggest adding "Healthy Adults", as the study is not targeting pathological population. 

2) Please provide brief discussion how frontopolar and lateral PFC are part of the striatal reward circuit. 

Round 2

Reviewer 1 Report

Comments and Suggestions for Authors

I thank the Authors for addressing my concerns. I have still a concern regarding the DPF. Maybe my question was misguiding, but I did not mean that the definition of the DPF is an issue for the comparisons proposed by the Authors. Simply, the DPF should be set a priori (or modeled/estimated as proposed by the following references) in order to compute oxy and deoxy hemoglobin. Hence the DPF should be defined in this study, maybe it is set as input of the device. Please check and report in the manuscript the DPF chosen.

Chiarelli, A. M., Perpetuini, D., Filippini, C., Cardone, D., & Merla, A. (2019). Differential pathlength factor in continuous wave functional near-infrared spectroscopy: Reducing hemoglobin’s cross talk in high-density recordings. Neurophotonics6(3), 035005-035005.

Scholkmann, F., & Wolf, M. (2013). General equation for the differential pathlength factor of the frontal human head depending on wavelength and age. Journal of biomedical optics18(10), 105004-105004.
